# Comparative Study on Using Various Recovery Stimulation Methods to Boost Nitrification Recovery in SBRs Inhibited by Hazardous Events

**Hoang-Duy Nguyen [1], Chih-Chi Yang [1],*, Khanh-Chau Dao [1,2] , Van-Phat Le [1] and Yung-Pin Tsai [1],***

1. Department of Civil Engineering, National Chi Nan University, Puli, Nantou 54561, Taiwan; s108322515@mail1.ncnu.edu.tw (H.-D.N.); daokhanhchau07@gmail.com (K.-C.D.); phatle2601@gmail.com (V.-P.L.)
2. Department of Health and Applied Sciences, Dong Nai Technology University, Bien Hoa 810000, Dong Nai, Vietnam
* Correspondence: yangcc@mail.ncnu.edu.tw (C.-C.Y.); yptsai@ncnu.edu.tw (Y.-P.T.)

**Abstract:** A system consisting of six SBR units was operated in parallel for three phases to investigate the impacts of salinity shock and anaerobic and aerobic starvation on the activated sludge process stability and effects of various recovery stimulation methods on the subsequent recovery period. Different recovery strategies were applied in each SBR unit, including natural recovery, adding bio-accelerators, a stepwise increase feed strategy, a stepwise strategy coupled with bio-accelerators dosing, extended aeration time, and extended aeration time coupled with bio-accelerators dosing. It was concluded that the combination of stepwise strategy and dosing bio-accelerators showed the most efficiency in boosting system recovery after being subjected to NaCl shock and starvation. The boosting effect of the stepwise strategy alone was slightly better in recovery after NaCl shock. Furthermore, extending the aeration rate could bring more positive effects when resuscitating the system after long-term anaerobic starvation. For the unit that only received dosing of bio-accelerators during the recovery period, it could be concluded that there was a specific time requirement for the bio-accelerators to take effect significantly, as the impact of bio-accelerators on the beginning days of recovery periods was very slight. In contrast, adjusting operational regimes such as stepwise increased feed volume or extending aeration time could significantly boost the SBRs from the first recovery days. Hence, highly effective recovery efficiency could be achieved by coupling dosing bio-accelerators with other operational adjustment methods, especially stepwise strategies.

**Keywords:** salinity shock; starvation; recovery stimulating; bio-accelerators

## 1. Introduction

In water and wastewater treatment, there are strict regulations imposed on the quality of effluent water applied by most regions in the world. Along with the advancement of environmental science and technology, those regulations are expected to become more stringent to lighten the burden placed on the surrounding environment of the receiving point. In that context, high stability is required in water/wastewater treatment units to maintain the overall process's high treatment efficiency. Moreover, in some developed countries such as Australia or America, implementing the wastewater treatment process for recycling requires validation or verification to test whether the technology can achieve the required water quality [1]. In the validation process, hazardous events (or 'upset conditions' as used in the US) are critical for examining the process's robustness in case of undesirable incidents. The US EPA requires records of upset condition occurrences in its verification report protocol [2]. Hazardous events such as sudden changes in inlet water composition, extreme weather events, operational error, and mechanical malfunctions can negatively affect the overall treatment efficiency of the technology [1], posing a potential risk of

violating the effluent discharge standard and deteriorating the surrounding environment of the receiving water body.

Among the hazardous events, salinity shock and starvation could easily be encountered in many wastewater treatment plants worldwide. Severe infiltration of seawater due to sewer leakage and high tide led to the recording of a substantial rise in the salt levels in an aeration tank [3]. Excess concentration of salinity in the activated sludge may cause some undesired issues to the treatment efficiency of the biological process, such as activity inhibition of ammonia-oxidizing bacteria (AOB) and nitrite-oxidizing bacteria (NOB) [4–7]. Similarly, several incidents could lead to long-term starvation of the activated process, such as power loss due to natural disasters, lockdown due to a severe pandemic, long-term holidays, etc. Some negative impacts including a loss in AOB and NOB activities [8,9], loss of biomass, and the appearance of opportunistic microorganisms [10] have been reported for systems subjected to starvation. As the nitrifiers are autotrophs and require a longer growth time than heterotrophs responsible for COD removal or denitrification, those hazardous events may have more pronounced effects on the nitrification process.

Until now, the number of research studies focusing on methods for boosting the recovery rate of the inhibited systems has been scarce. No reports about recovery stimulation methods for salinity-inhibited activated sludge systems could be found. However, some methods for boosting the recovery rate of starved biomass, such as stepwise load strategy [11] or nitrogen loading rate control loop [12], have been studied with promising results observed. Recently, a group of chemicals known as bio-accelerators, e.g., biotin, 6-benzylaminopurine, and L-aspartic acid, have been found to have a considerable stimulating effect on recovery in heavy metal and ammonium-inhibited systems [13–15]. Hence, dosing bio-accelerators could be a potential method for boosting the recovery rate of inhibited activated sludge.

Furthermore, integrated methods related to the introduction of bio-accelerators and a stepwise strategy or lengthening of aerobic reaction time could also be a great option for shortening the time required for system recovery after upset conditions. Therefore, this research on "enhancing process recovery via modifying operational condition coupling with dosing bio-accelerators in SBRs inhibited by hazardous events" was conducted to elucidate the issues above. Specifically, the impact of adopting recovery stimulation methods such as adjusting operational regime or adding bio-accelerators, both separately and in combination, on recovery of nitrifiers in the SBRs were studied and compared.

## 2. Materials and Methods

### 2.1. Inoculum-Activated Sludge and Synthetic Wastewater

The inoculum-activated sludge utilized in this study was taken from the sequencing batch reactor (SBR) of the wastewater treatment system on the National Chi Nan University campus. The daily flow rate of this plant is 700 $m^3$/day, and the biomass concentration was 5430 mg/L as mixed liquor suspended solid (MLSS) and 4350 mg/L as mixed liquor volatile suspended solid (MLVSS) at the time of sampling. Hence, the MLVSS/MLSS ratio of the raw sludge sample was roughly 0.8, indicating good sludge characteristics. After the raw activated sludge had been collected, it was sieved through a 0.5 mm sieve to remove undesired particles before pouring into the 12 L cultivation tank in the laboratory. Then, the sludge was operated in the cultivation tank for one month before being evenly distributed into the experimental units.

The following chemicals were dissolved in reverse osmosis (RO) water in order to create synthetic wastewater: $NH_4Cl$ (160 mg/L); glucose (281.25 mg/L); $NaHCO_3$ (502 mg/L); $KH_2PO_4$ (5 mg/L); $Na_2HPO_4$ (20 mg/L). This resulted in roughly 40 mg/L of ammonium nitrogen ($NH_4^+$_N), 5 mg/L of phosphate ($PO_4^{3-}$_P), and 300 mg/L of chemical oxygen demand (COD) present in the synthetic wastewater. This type of synthetic wastewater represented mid-strength municipal wastewater and was used in both the initial cultivation period mentioned above and for the operation of the SBRs in the whole study. Bio-accelerators (BAs), i.e., biotin, 6-benzylaminopurine, and L-aspartic acid, were prepared separately as

500 mL stock solutions at the concentration of 0.1 mg/L. When preparing stock solution biotin or 6-benzylaminopurine, drops of NaOH 1M were gradually added to the mixing solution until those powdered chemicals dissolved completely into the liquid phase.

## 2.2. Experimental System and Experimental Procedures

In this research, 6 identical SBRs, as regarded henceforth as SBR 1, SBR 1′, SBR 2, SBR 2′, SBR 3, SBR 3′, were used to study the effects of different stimulation methods on the process recovery. The configuration of the SBR system is shown in Figure 1 below:

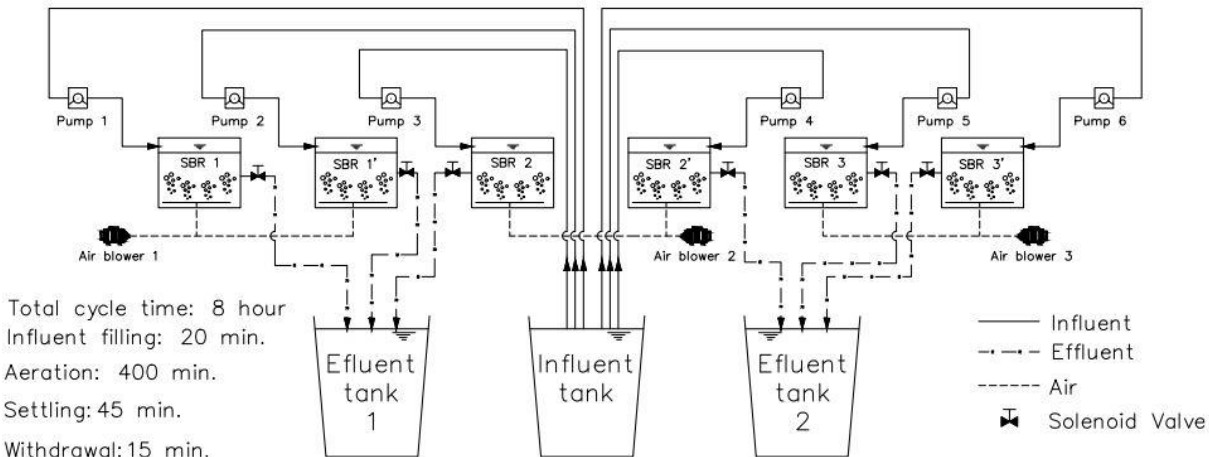

**Figure 1.** SBR system configuration.

The SBRs had a working volume of 2 L for each unit and were simultaneously operated for 3 cycles per day. For each cycle, 0.8 L of synthetic wastewater was treated during each 8 h cycle (20 min of influent filling, 400 min of aeration, 45 min of settling, and 15 min of effluent withdrawal) in the SBR exchange ratio of 0.4. Hence, the hydraulic retention time (HRT) of each reactor was 20 h. During the study period, the mixed liquor was wasted periodically to maintain the inner MLSS concentration of the lab-scale SBRs at about 4000 mg/L. The temperature and pH of the mixed liquors were maintained at $20 \pm 2\,^{\circ}\mathrm{C}$ and $7.6 \pm 0.2$. The SBRs entered the research period when high COD treatment and nitrification efficiency were achieved and maintained for at least 1 week.

## 2.3. Experimental Set-Up

For conducting the experiment in several types of hazardous events, the study was divided into 3 phases, named by the hazardous events studied in the phases: salinity shock, anaerobic starvation, and aerobic starvation. Specifically, 30 g/L of NaCl was added to the synthetic wastewater for conducting a salinity shock experiment, also regarded as Phase 1. After 3 consecutive phases treating wastewater containing NaCl, the SBR system entered the recovery phase. The starvation condition was studied under anaerobic and aerobic regimes, subsequently regarded as Phases 2 and 3. The length of starvation was 14 days in both regimes, and no influent wastewater was introduced into the system during the time. During aerobic starvation, aeration was continuously supplied, while there was no air supplied to the SBRs during anaerobic starvation, leading to the biomass sedimentation to the tanks' bottom. Before taking samples to measure the mixture's change during anaerobic starvation, the biomass was temporarily mixed for 2 min. Like the salinity shock phase, the SBRs entered recovery after going through 14 days of anaerobic starvation in Phase 2 and aerobic starvation in Phase 3.

The SBR 1 served as a control system that went through a natural recovery process after being subjected to the upset conditions. The recovery efficiencies of other SBRs were compared with the control reactor's recovery behavior to evaluate the boosting effects by artificial methods towards recovery. Adjustments in operational conditions for boosting

process recovery were applied on SBR 2 and SBR 3. For SBR 2, specifically, the recovery phase after terminating the hazardous event period was divided into 3 sub-stages, i.e., sub-stage 1, 2, and 3, corresponding with the inlet volume of 50%, 75%, and 100% compared with the initial value. In other words, SBR 2 began its recovery stage under the regime of sub-stage 1 and gradually transferred to sub-stage 2 and 3 when a desirable nitrification rate had been reached. Similarly, the aeration time in each SBR cycle of SBR 3 changed from 400 min to 640 min after ceasing hazardous event experiments, leading to a rise in cycle time from 8 h initially to 12 h. Hence, there were only 2 cycles per day. This also divided the recovery period of the SBR 3 into 2 sub-stages. After reaching a nitrification rate that was as high as observed in the steady-state, the aeration time of SBR 3 was adjusted back to the initial value. In SBR 1', SBR 2', and SBR 3', bio-accelerators (BAs) were added to the mixed liquor throughout the recovery period. At the same time, the operational regimes of those units were similar to those of SBR 1, SBR 2, and SBR 3, respectively. At the beginning of each SBR cycle during the recovery period (i.e., after influent had been filled and aeration began), 1 mL from each BA stock solution was directly added to SBR 1', SBR 2', and SBR 3', resulting in the presence of 0.1 µg/L of each BA in the mixed liquors.

*2.4. Analytical Methods*

Parameters including MLSS, MLVSS, COD, $NH_4^+\_N$, $NO_2^-\_N$, $NO_3^-\_N$, $PO_4^{3-}\_P$ were analyzed following the standard methods (APHA, AWWA, WEF, 2017) [16] by two parallel samples. The pH of the mixed liquor was measured by EZDO pH meter (GOnDO, Taiwan).

Specific oxygen uptake rate (SOUR) of AOB and NOB were analyzed as per the method introduced by Surmacz-Gorska et al. [17], with some modification. Specifically, $SOUR_{AOB}$ and $SOUR_{NOB}$ stand for the SOUR of AOB and NOB, respectively. For each experiment, 100 mL of the mixed liquor from each SBR was sampled for measuring the bacterial SOUR. $NH_4^+\_N$ and $NO_2^-\_N$ with concentrations of 40 mg/L and 20 mg/L, respectively, were altogether provided, and the mixed liquor was aerated for 1 h before measurements of dissolved oxygen (DO) concentration. Drops of NaOH 0.1M were also added to prevent a drop in pH value during the experiment. After 1 h of aeration, the mixed liquor was fed into a closed beaker with agitation for DO measurements. The initial DO concentration when terminating the aeration step was about 8 mg/L, and pH was controlled at 7.5 ± 0.2. Agitation of 90 rpm was provided during the measurement by a magnetic stirrer system.

Before *DO* measurements, *MLSS* and *MLVSS* of the mixed liquor sample were also analyzed for subsequent *SOUR* calculation. *SOUR* values were calculated as per the following concept:

$$SOUR\ (mgO_2/g_{MLVSS}.min) = \frac{DO\ utilised\ by\ the\ bacteria\ (mgO_2/L.min)}{MLVSS\ (g/L)} \quad (1)$$

In the *DO* measurement step, the *DO* concentration of the sample was monitored continuously. The *SOUR* recorded in this stage was regarded as total *OUR* and was the average value measured in 4 min (from 0:00 to 4:00). At the 4th minute, *NaClO₃* with a concentration of 20 mM or 2.13 g/L was added to the sample, and *DO* concentration was continued to be measured for another 4 min (from 4:00 to 8:00). As *NaClO₃* is an inhibitor of $NO_2^-$ oxidation by Nitrobacter at a concentration of 20 mM, $SOUR_{NOB}$ is the difference between the total *OUR* and the *OUR* measured in the presence of *NaClO₃*. In specific, the $SOUR_{NOB}$ was calculated as:

$$SOUR_{NOB}\ (mgO_2/g_{MLVSS}.min) = \frac{OUR_{total}(mgO_2/L.min) - OUR_{NaClO_3}(mgO_2/L.min)}{MLVSS\ (g/L)} \quad (2)$$

After the experiment was continued for another 4 min, *ATU* with a concentration of 5 mg/L was added to the sample. The *DO* was continued to be measured for 4 min with both *NaClO₃* and *ATU* (from 8:00 to 12:00). As *ATU* can inhibit $NH_4^+$ oxidation by Nitrosomonas at the concentration of 5 mg/L, SOURAOB can be determined from the

difference between *OUR* with $NaClO_3$ and *OUR* with both $NaClO_3$ and *ATU*. In specific, the $SOUR_{AOB}$ was calculated as:

$$SOUR_{AOB}(mgO_2/g_{MLVSS}.min) = \frac{OUR_{NaClO_3}(mgO_2/L.min) - OUR_{NaClO_3+ATU}(mgO_2/L.min)}{MLVSS\ (g/L)} \qquad (3)$$

The measured *SOUR* figures were used to examine the microorganisms' activity during steady-state, after being subjected to upset conditions, and during recovery. From the difference in *SOUR* values between steady-state and after hazardous events as well as the recovery period, the inhibition rate could be calculated using the following equation:

$$Inhibition\ rate\ (\%) = \frac{SOUR_S - SOUR_H}{SOUR_S} \times 100\% \qquad (4)$$

in which:

$SOUR_S$: $SOUR_{AOB}$, $SOUR_{NOB}$ values of the mixed liquor during the steady-state;
$SOUR_H$: $SOUR_{AOB}$, $SOUR_{NOB}$ of the mixed liquor after the hazardous events;

During the recovery period, *SOUR* values and substrate removal efficiency were equally used to evaluate the recovery efficiencies of each SBR system.

## 3. Results and Discussion

### 3.1. Salinity Shock

Figure 2 shows the concentration of $NH_4^+$_N, $NO_2^-$_N, and $NO_3^-$_N in the effluent, while Figure 3 provides data about the SOUR value of AOB and NOB of the SBRs. As shown above, in Figure 2a,b, the concentration of $NH_4^+$_N and $NO_2^-$_N in the effluent wastewater in all SBRs was close to zero, indicating very high nitrification efficiency. The concentrations of $NO_3^-$_N (Figure 2c) were kept in the range of 23.31 and 27.37 mg/L throughout the steady-state operation period. Average values of $SOUR_{AOB}$ and $SOUR_{NOB}$ of the SBRs values of the SBRs were measured to be $0.106 \pm 0.008$ and $0.0304 \pm 7 \times 10^{-4}$ $mgO_2$/gMLVSS.min, respectively.

After NaCl was introduced into the SBRs on operation day 7, the average NH4+_N and $NO_2^-$_N concentration in the SBR effluent rose to $8.4 \pm 1.05$ mg/L and $11.03 \pm 0.8$ mg/L, respectively, while those of $NO_3^-$_N dropped to $8.45 \pm 0.9$ mg/L. Correspondingly, the average $SOUR_{AOB}$ and $SOUR_{NOB}$ in the SBRs also dropped to $0.0187 \pm 0.0016$ and $0.0104 \pm 0.0012$ $mgO_2$/gMLVSS.min, respectively, and the corresponding inhibition rates were $82.4 \pm 1.78\%$ and $65.78 \pm 4.13\%$. Those experimental results indicate that both Nitrosomonas and Nitrobacter group were severely affected by the presence of NaCl, as was previously reported in many studies [4,18]. Similar to this study, the MBR model of Yogalakshmi et al. [18] also recorded about 80% removal efficiency of TKN at NaCl dosage of 30 g/L, while the $A_2O$ model of Panswad et al. [7] only demonstrated total nitrogen removal rate at 70% at the same NaCl concentration, even after the model had reached steady state with salinity-acclimatized sludge. They noticed that the inlet TN concentration used by Panswad et al. [7] was merely 25 mg/L compared with Yogalakshmi et al. [19] with 134 mg/L of TKN and this research with 40 mg/L of $NH_4^+$_N. Hence, it is clear that with high biomass concentration, i.e., 12 g/L of MLSS at 30 gNaCl/L in Yogalakshmi et al. [19] and 4000 mg/L of MLSS in this study, the ability for nitrogen removal under high salinity concentration could also be kept around 80%, while the initial biomass concentration of 2500 mg/L in the $A_2O$ model of Panswad et al. [7] could not help the system to reach desirable nitrogen removal efficiency under the same condition. In other words, the biological systems operated at denser biomass concentrations can show higher tolerability toward salinity shock than other systems cultivating lower biomass concentrations. However, only a 20% removal rate of TKN reported by Yogalakshmi et al. [19] at 70 g/L salinity shock indicated that the salt tolerability of biomass does not increase linearly with the MLSS concentration.

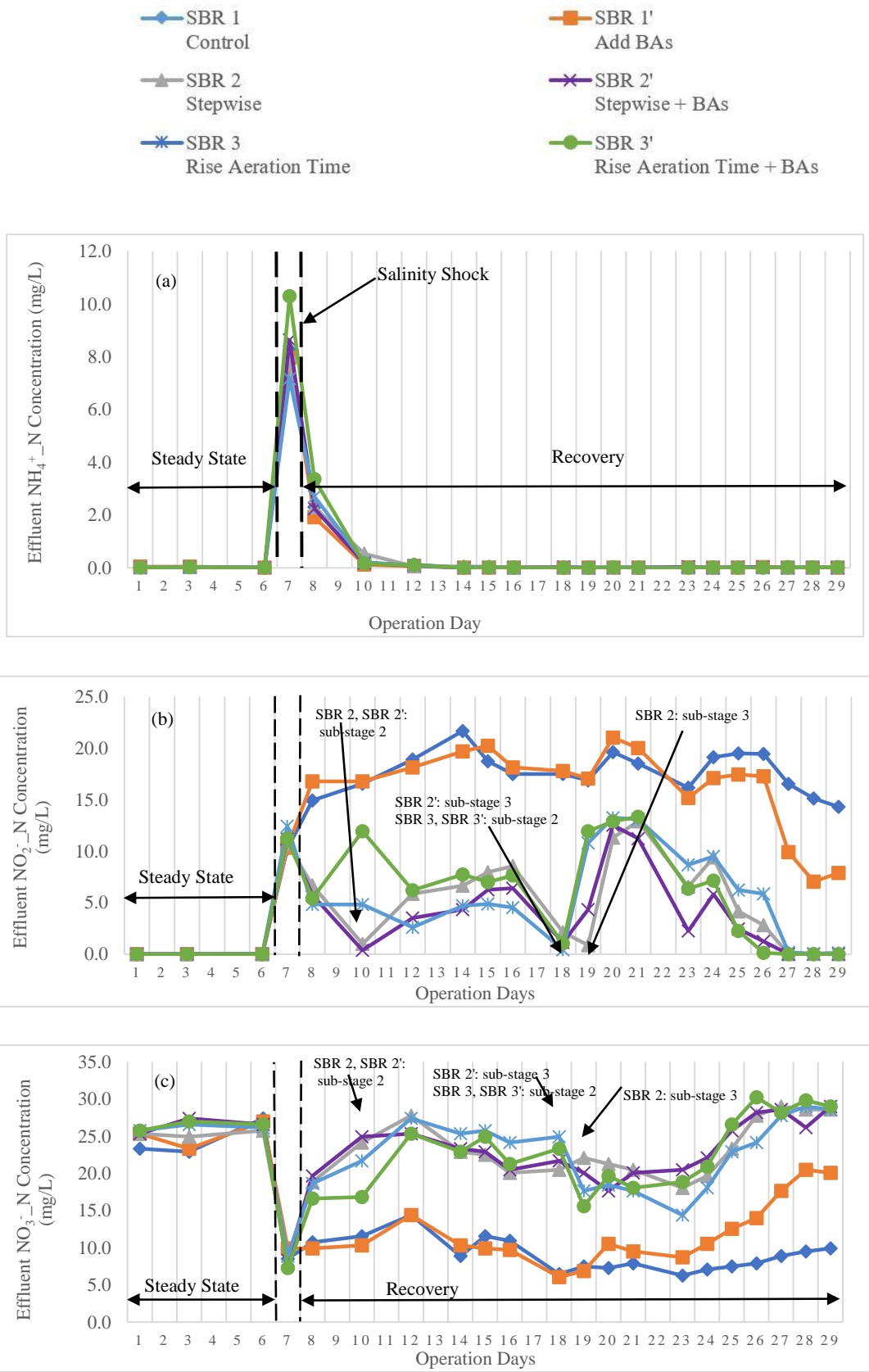

**Figure 2.** Phase 1—Effluent $NH_4^+$_N concentration (**a**), $NO_2^-$_N concentration (**b**), $NO_3^-$_N concentration (**c**).

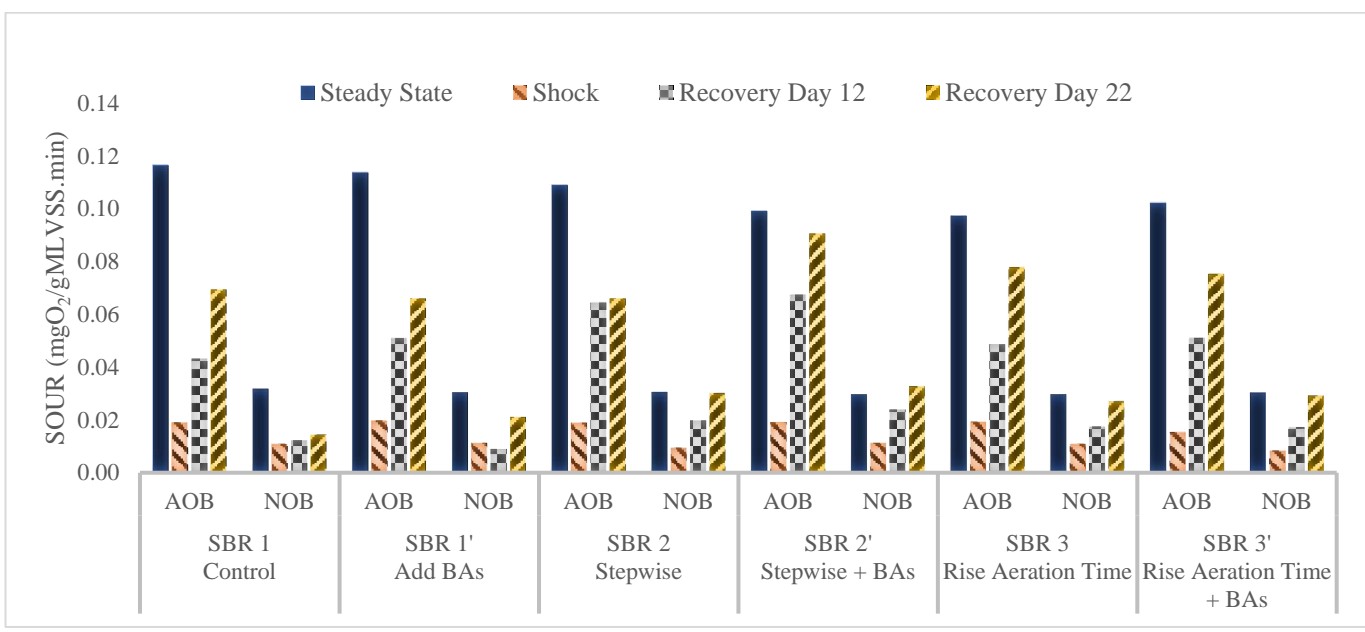

**Figure 3.** Phase 1—SOUR of AOB and NOB.

As the Nitrosomonas genus is halotolerant [20], the ammonium nitrogen removal rate of all SBRs in the system recovered quickly after ceasing NaCl (Figure 2a). On the third recovery day, the effluent NH4+_N reached $0.23 \pm 0.16$ mg/L, equivalent to $99.5 \pm 0.36\%$ of $NH_4^+$_N removal efficiency. The experimental results from the seventh recovery day (operation day 14) until the end of Phase 1 revealed that the $NH_4^+$_N removal rate of all SBRs continued to be stably high until the end of the recovery period (operation day 29). The SOUR$_{AOB}$ figures in SBR 1 to SBR 3′ were measured to be 0.0432, 0.0510, 0.0645, 0.0675, 0.0486, 0.0512 mgO$_2$/gMLVSS.min, respectively (Figure 3), on the 12th recovery day. It could be observed that the recovery rate of SBR 2 and SBR 2′ with stepwise loading strategy showed better effects in AOB activity recovery. At the beginning of the recovery period, the biomass in SBR 2 and SBR 2′ was first compacted in the total mixed liquor volume of 1.6 L when only 50% of the initial feed was introduced. Hence, creating denser MLSS concentrations could provide higher tolerance and recovery rate during shock loading inhibition. Meanwhile, increasing aeration provided little change because SBR 1 and SBR 1′ had already reached as high as 99% $NH_4^+$_N removal rate without extra aeration time. At the end of the recovery phase, the SOUR$_{AOB}$ figures in the SBRs were 0.0695, 0.0661, 0.0660, 0.0907, 0.0779, 0.0753 mgO$_2$/gMLVSS.min, respectively. In the latter period of the recovery phase, it was expected that there would be a better $NH_4^+$_N oxidation rate in other SBRs as compared with SBR 1. However, the SOUR$_{AOB}$ value of 0.0660 mgO$_2$/gMLVSS.min in SBR 2 was the lowest among the SBRs, while the SBR 2′, SBR 3, and SBR 3′ recovery rates were considerably higher than SBR 1 and SBR 1′. The reason for this phenomenon is unknown.

The recovery of NOB activity after salinity shock in this research was a lot more complicated than that of AOB activity (Figure 2b,c). Considering the effluent $NO2^-$_N concentration, SBR 3′ reached 0.01 mg/L of $NO_2^-$_N on the 19th recovery day (operation day 26) and was the fastest in this phase. At the same time, both SBR 2 and SBR 2′ needed 20 days for achieving the same efficiency. On the 22nd recovery day (operation day 29), SBR 2′ reached 0 mg/L of $NO_2^-$_N in the effluent, which was the most efficient result in this phase. SBR 3 reached as low as 0.18 mg/L of $NO_2^-$_N concentration in the effluent on the 20th recovery day but later on fluctuated between 0.05 and 0.14 mg/L. At the end of the recovery period, the $NO2^-$_N concentrations in SBR 1 and SBR 1′ were 15.11 and 7.04 mg/L, respectively, and $NO_3^-$_N concentrations in SBR 1 and SBR 1′ were 9.5 and 20.47 mg/L, respectively. Although SBR 1′ showed a considerably better recovery boosting

effect than SBR 1, both reactors failed to achieve the initial nitrification efficiency as recorded before the occurrence of salinity shock.

From recovery day 1 to day 14 (operation day 8 to 21) of the recovery period, the $NO_2^-$_N concentration of SBR 1′ was nearly equal and sometimes even higher than that in SBR 1 (Figure 2b). From recovery day 16 to day 19 (operation day 23 to 26), there was a gradual decrease om $NO_2^-$_N concentration coupled with a considerable rise in $NO_3^-$_N concentration. From recovery day 20 (operation day 27) to the end of the recovery phase, the difference between nitrite and nitrate nitrogen in the effluent of SBR 1 and SBR 1′ became more evident, as shown in Figure 2b. The $SOUR_{NOB}$ values here in SBR 1 and SBR 1′ were 0.0145 and 0.0212 $mgO_2$/gMLVSS.min (Figure 3), respectively, further confirming the higher recovery effectiveness in SBR 1′ with bio-accelerators addition compared with SBR 1 without dosing bio-accelerators.

The advantages of the low carbonaceous and ammonium shock load of stepwise strategy enabled SBR 2 and SBR 2′ to achieve highly efficient nitrification quickly. Both reactors were simultaneously transferred to sub-stage 2 after only 3 recovery days (Figure 2b). SBR 2 then needed 9 days for reaching sub-stage 3, while the SBR 2′ only took 8 days to achieve that target. Both SBRs reached an effluent $NO_2^-$_N concentration of 0.01 mg/L on the 20th recovery day (operation day 27). The effluent $NO_2^-$_N concentration of SBR 2′ on the 22nd recovery day (operation day 29) reached 0 mg/L, while SBR 2 fluctuated around 0.01 and 0.02 mg/L. The corresponding $SOUR_{NOB}$ values of SBR 2 and SBR 2′ were 0.0303 and 0.0328 $mgO_2$/gMLVSS.min, respectively (Figure 3). As the SBR 2′ unit illustrated slightly better nitrification efficiency than SBR 2 and was utterly outstanding compared with SBR 1 and SBR 1′, it can be concluded that the method of stepwise strategy had significantly boosted the recovery process of the salinity-inhibited SBRs. In addition, the dosing of bio-accelerator compounds into the biomass helped polish the quality of effluent wastewater; hence, the effect achieved with bio-accelerators should not be neglected.

Although extending aeration time could also lower the total carbonaceous and ammonium loading rate per day, both SBR 3 and SBR 3′ went through 11 operation days at sub-stage 1 before transferring to sub-stage 2 (Figure 2b). SBR 3′ then reached $NO_2$-_N concentration of 0.12 mg/L on the 19th recovery day (operation day 26) and was maintained at 0.01 mg/L until the last day, while SBR 3 reached 0.18 mg/L $NO_2^-$_N on the 20th recovery day (operation day 27) but then fluctuated between 0.05 and 0.14 mg/L in the last 2 days, indicating a slight unstable of $NO_2^-$_N oxidation efficiency of NOB without the addition of bio-accelerators. The corresponding $SOUR_{NOB}$ values of SBR 3 and SBR 3′ were 0.0273 and 0.0293 $mgO_2$/gMLVSS.min, respectively (Figure 3).

Through observation in a fluctuation of effluent $NO_2^-$_N concentration of all six SBRs, it is evident that SBR 3′ recovered within 19 days and hence was the fastest in the phase compared with SBR 2 and SBR 2′, which both needed 20 days to reach 0.01 mg/L $NO_2^-$_N. However, only SBR 2′ reached 0 mg/L of $NO_2^-$_N in the effluent on the last day and the $SOUR_{NOB}$ value of the SBR 2′ was also the highest at the end of the recovery phase (0.0328 $mgO_2$/gMLVSS.min). Hence, the stepwise loading strategy with the bio-acceleration introduction was the most efficient recovery boosting method among the methods proposed in this study.

Cell plasmolysis occurs under sudden salt shock conditions, causing the loss of water molecules within microbial cells and subsequent inhibition of microbial growth or cell death. Furthermore, high salinity can also interfere with the metabolism of microorganisms and cause damage to the microbial cell membrane and enzymes [21]. Considering the composition of BAs used in this study, cytokinin can help boost cell division, while dosing biotin can mitigate the loss of membrane integrity. On the other hand, L-aspartic acid is an indispensable element for cell growth. It is a linear chain structure containing an amide bond (N–H) that microorganisms can quickly assimilate.

Furthermore, ammonia can also be produced by L-aspartic acid through the deamination process. Wang et al. [13] reported that L-aspartic acid was more effective in boosting process recovery than biotin and cytokinin. However, as the $NH4^+$_N source from deami-

nation and N–H bond in L-aspartic acid structure seemed more favorable for the growth of AOB rather than NOB, the recovery stimulating effect of L-aspartic acid, in this case, was not essential. Furthermore, NOB has less ability to tolerate a high salinity environment than the AOB genus. Therefore, there was a high possibility that the death of microorganisms took a large proportion in the activity reduction measured as $SOUR_{NOB}$. Autotrophic NOB may also require a longer time, even with BAs dosage, for complete recovery due to its low growth rate, which is even lower than AOB (0.1–0.15 g VSS/g $NH_4^+$-N for AOB compared with 0.04–0.07 g VSS/g $NO_2^-$-N for NOB [22]), as mentioned above. In contrast, adopting operational condition adjustment methods can reduce the substrate loading rate in the SBR units; hence, the accumulation rate of nitrite, a well-known inhibitor of nitrifiers, was significantly reduced. Therefore, for recovery of NOB genus after the inhibition by salinity shock, adjusting the operational condition would have a more stimulating effect toward process recovery than bio-accelerator dosing.

### 3.2. Anaerobic Starvation

As shown in Figure 4 below, the AOB and NOB genera activities were degraded to some extent. The average $SOUR_{AOB}$ of the SBRs dropped from an initial $0.0640 \pm 0.003$ mgO$_2$/gMLVSS.min to $0.038 \pm 0.003$ mgO$_2$/gMLVSS.min after 14 days of anaerobic starvation, corresponding to an inhibition rate of $40.22 \pm 6.5\%$. For the NOB genus, the $SOUR_{NOB}$ dropped from the initial value of $0.0269 \pm 0.0026$ mgO$_2$/gMLVSS.min to $0.0185 \pm 0.0013$ mgO$_2$/gMLVSS.min, corresponding to an inhibition rate of $30.82 \pm 5.2\%$, respectively. There was a significant drop in pH (Supplementary Materials Figure S1) caused by the release of H$^+$ ions to the liquid phase due to the nitrite oxidation process (Anthonisen et al. 1976) [22]. The chemical reaction of the process can be presented as below:

$$NH_4 + 1.5\,O_2 \rightarrow H_2O + H^+ + NO_2^- \rightleftharpoons HNO_2$$

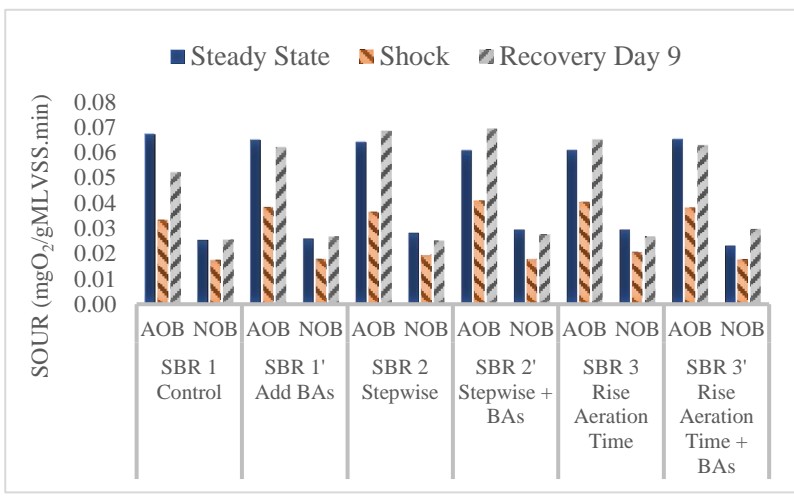

**Figure 4.** Phase 2—SOUR of AOB and NOB during the experiment.

In this case, it should be noticed that the presence of $NO_2^-$\_N coupling with the pH drop in the starting period might have increased free nitrous acid HNO$_2$, an inhibitory substance to both AOB and NOB [22]. Additionally, the drop in activity of AOB and NOB was also possibly caused by cell decay that had been evidenced by the rise of ammonium and phosphate concentration in the medium (Supplementary Materials Figure S1). Ma et al. [8] also observed deterioration of activity of nitrifiers with a specific decay rate at 0.017 d$^{-1}$ for AOB and 0.029 d$^{-1}$ for NOB under anoxic–anaerobic starvation conditions over 25 days. Other data relating to the changes in other parameters, including COD,

$NH_4^+$_N, $NO_2^-$_N, $NO_3^-$_N, $PO_4^{3-}$_P, MLSS, and MLVSS during the starvation period, can be found in the Supplementary Materials Figures S1 and S2.

The effluent $NH_4^+$_N, $NO_2^-$_N, and $NO_3^-$_N concentrations of the SBRs during the recovery period are illustrated in Figure 5a–c, respectively. At first glance, it is evident that there had been a significant fluctuation in nitrification recovery efficiency in different SBRs.

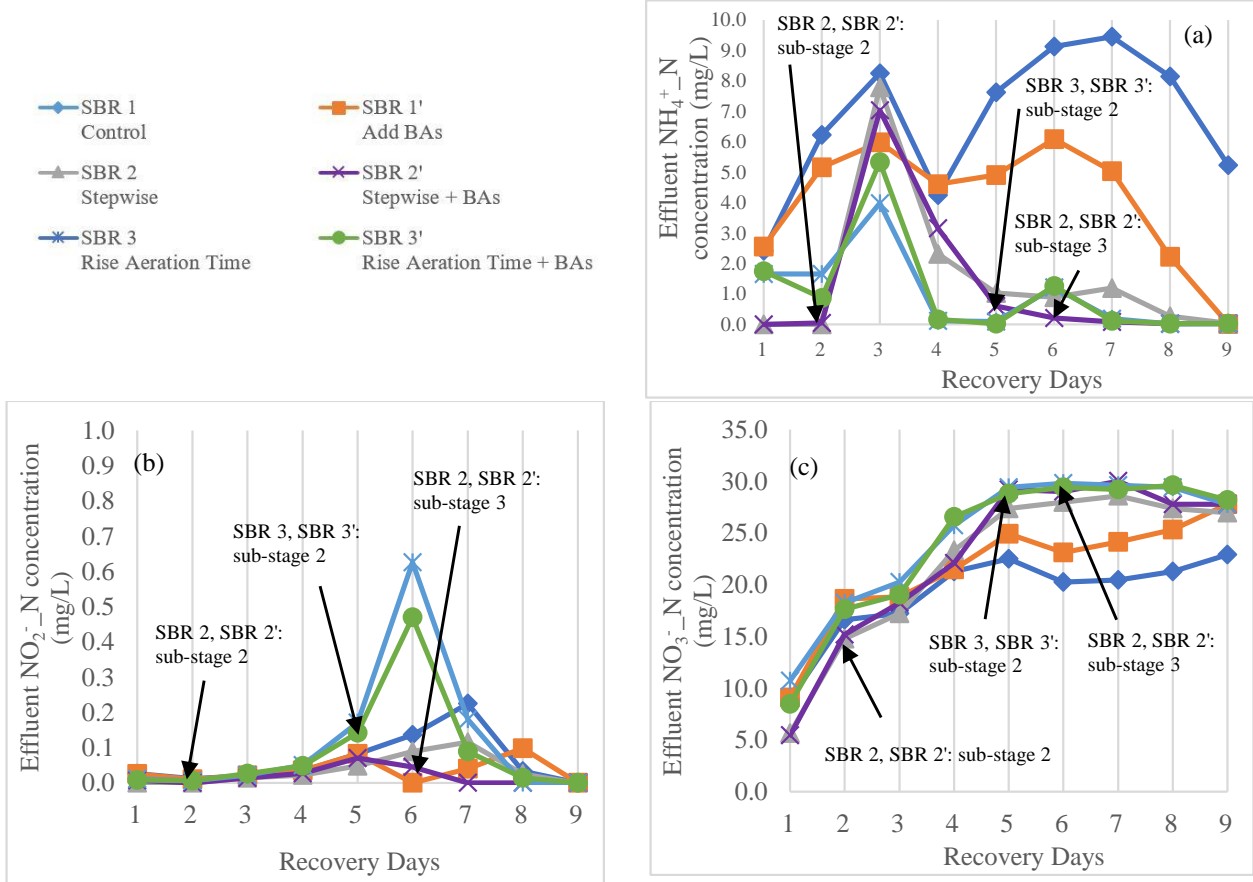

**Figure 5.** Phase 2—Effluent $NH_4+$_N concentration (**a**), $NO_2^-$_N concentration (**b**), $NO_3^-$_N concentration (**c**).

For SBR 1 and SBR 1′, $SOUR_{AOB}$ deterioration due to anaerobic starvation made the nitrifiers in the systems unable to completely oxidize the ammonium nitrogen present in the effluent. This led to the subsequent $NH_4^+$_N accumulation in the mixed liquor, as can be observed by rising trends in the effluent $NH_4^+$_N concentration. During the recovery period, the concentration of $NH_4^+$_N measured in the effluent of SBR 1 fluctuated largely from 2.42 to 9.45 mg/L. On the ninth recovery day, the effluent $NH_4^+$_N concentration of SBR 1 was 5.23 mg/L, corresponding to an $NH_4^+$_N removal efficiency of 86.08%. For SBR 1′, the $NH_4^+$_N concentration measured in the effluent also significantly fluctuated until the sixth recovery day before dropping dramatically, and only 0.02 mg/L of $NH_4^+$_N was measured on the ninth recovery day. Hence, the $NH_4^+$_N removal efficiency of SBR 1′ reached 99.95%, and the $NO_3^-$_N concentration measured was 27.78 mg/L. Proper NOB activity was evidenced by a negligible concentration of $NO_2^-$_N recorded in the effluent of both SBR 1 and SBR 1′ during the recovery period. On the ninth recovery day, the SOUR analysis revealed that the $SOUR_{AOB}$ values of SBR 1 and SBR 1′ were 0.0521 and 0.0620 $mgO_2$/gMLVSS.min, respectively, and the $SOUR_{NOB}$ values of SBR 1 and SBR 1′ were 0.0255 and 0.0269 $mgO_2$/gMLVSS.min, respectively (Figure 4).

Similar to Phase 1, both SBR 2 and SBR 2′ reached very high nitrification efficiency after the first days of the recovery period. Both reactors were transferred to sub-stage

2 on the second recovery day before reaching sub-stage 3 on the sixth recovery day 6. The $NH_4^+\_N$ concentration measured in the effluent of SBR 2′ continued to drop to 0.08 mg/L despite the rise in the amount of $NH_4^+\_N$ to be oxidized by increasing the feed flow rate. At the same time, the effluent $NH_4^+\_N$ concentration in SBR 2 rose slightly to 1.2 mg/L. The effluent $NH_4^+\_N$ concentration of SBR 2 then dropped to 0.03 mg/L on the ninth recovery day. The $NH_4^+\_N$ removal efficiency values of SBR 2 and SBR 2′ on the last recovery day were 99.91 and 99.94%, respectively. On the seventh recovery day, SBR 2′ was considered fully recovered, with 99.78% $NH_4^+\_N$ removal efficiency without nitrite accumulation. In contrast, SBR 2 was only considered recovered on the ninth recovery day due to some nitrite observed on the previous days. Correspondingly, the $SOUR_{AOB}$ values of SBR 2 and SBR 2′ were 0.0685 and 0.0693 $mgO_2$/gMLVSS.min, respectively, and the $SOUR_{NOB}$ values of SBR 2 and SBR 2′ were 0.0251 and 0.0277 $mgO_2$/gMLVSS.min, respectively (Figure 5).

At the beginning of the recovery period, the effluent $NH_4^+\_N$ values of SBR 3 and SBR 3′ were better than those of SBR 1 and SBR 1′ but were not as good as those from SBR 2 and SBR 2′. Both SBR 3 and SBR 3′ entered sub-stage 2 on the fifth recovery day. The $NH_4^+\_N$ concentrations measured in SBR 3 and SBR 3′ effluent were then slightly fluctuated before dropping to 0.02 on the eighth recovery day. However, a slight increase in $NO_2^-\_N$ concentration was recorded in the effluent of both SBR 3 and SBR 3′ on the following recovery days. Both SBR 3 and SBR 3′ were considered fully recovered on the eighth recovery day when the $NH_4^+\_N$ removal rate of both units reached 99.9% and $NO_2^-\_N$ concentration reached 0.01 mg/L (SBR 3′) and 0 mg/L (SBR 3). As different from the results recorded in SBR 1 and 1′ and SBR 2 and 2′, which showed the higher value of nitrifiers' SOUR in the unit with BAs addition, the $SOUR_{AOB}$ of SBR 3′ was lower than that of SBR 3 (0.0628 compared with and 0.0650 $mgO_2$/gMLVSS.min). In contrast, the $SOUR_{NOB}$ values of SBR 3 and SBR 3′ were 0.0269 and 0.0297 $mgO_2$/gMLVSS.min, respectively (Figure 5).

Except for SBR 1, which only reached 77.5% of the initial $SOUR_{AOB}$, the $SOUR_{AOB}$ and $SOUR_{NOB}$ value measured from other SBRs on the last recovery day was very close to those in steady-state or, in some cases, even higher than their initial value. For example, the $SOUR_{AOB}$ of SBR 2′ was the highest value for AOB activity in the system, while $SOUR_{NOB}$ measured in SBR 3′ was the highest for NOB activity. The days required for complete recovery in SBR 2′ were also the shortest (7 days), followed by SBR 3 and SBR 3′, which both fully recovered on the eighth recovery day, and SBR 1′ and SBR 2 both fully recovered on the ninth recovery day.

### 3.3. Aerobic Starvation

SOUR values of AOB and NOB in all SBRs were not affected by 14 starvation days, as shown in Figure 6. The $SOUR_{AOB}$ of the SBRs ranged from an initial $0.0535 \pm 0.0025$ $mgO_2$/gMLVSS.min to $0.0536 \pm 0.0039$ $mgO_2$/gMLVSS.min after 14 starvation days. Similarly, $SOUR_{NOB}$ of the SBRs also slightly fluctuated from $0.0251 \pm 7 \times 10^{-4}$ and $0.0256 \pm 0.0023$ $mgO_2$/gMLVSS.min before and after the starvation period, respectively. This result is contrary to almost all previous studies that observed activity deterioration to several extents caused by long-time starvation of nitrifying bacteria [9,11,12,23]. It could be explained that the nitrifiers of the SBRs in this case still had some ammonium to oxidize, as observed by the continuous increase in $NO_3^-\_N$ (Supplementary Materials Figure S3). This amount of ammonium could play a role in maintaining the activity of both AOB and NOB. Morgenroth et al. [24] reported little loss of nitrifying activities in 6 idle days due to the utilization of the ammonia released through the decay of heterotrophic bacteria to the liquid phase. Furthermore, as adequate alkalinity is also present in the medium for nitrification, the AOB and NOB in this study were not subjected to low pH inhibition, as found by Liu et al. [9]—hence avoiding hazardous risks that could restrict the nitrifying activities of the biomass. Other data relating to the changes in other parameters, including COD, $NH_4^+\_N$, $NO_2^-\_N$, $NO_3^-\_N$, $PO_4^{3-}\_P$, MLSS, and MLVSS during the starvation period, can be found in the Supplementary Materials Figures S3 and S4.

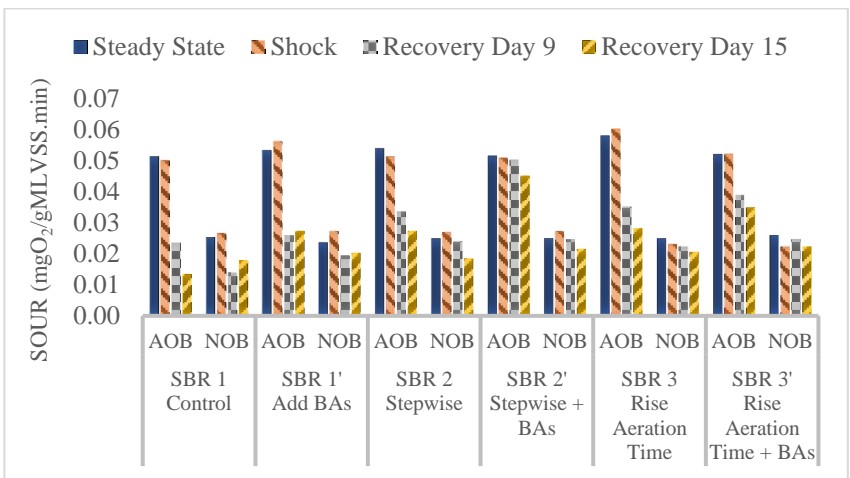

**Figure 6.** Phase 3—SOUR of AOB and NOB during the experiment.

In the first cycle after resuscitating the whole system, very effective nitrification was observed in the SBRs, with completely no ammonium and nitrite nitrogen detected in the effluent, except for SBR 1, with an effluent $NH_4^+\_N$ concentration of 0.41 mg/L. Hence, both SBR 2 and SBR 2′ were transferred to sub-stage 2 after merely three SBR cycles (equivalent to 1 operation day). The $NH_4^+\_N$ and $NO_2^-\_N$ concentrations of SBR 2, SBR 2′, SBR 3, and SBR 3′ were also negligible on day 2, and the operational regime of SBR 2 and 2′ and SBR 3 and 3′ were adjusted back to their initial condition. However, the $NH_4^+\_N$ concentrations of SBR 1 and SBR 1′ on day 2 rose to 1.74 and 1.04 mg/L, respectively, indicating the commencement of the deterioration in ammonium oxidation efficiency in the two SBRs. As shown in Figure 6a, the effluent concentration of $NH_4^+\_N$ in most SBRs increased over time. The precise reason or mechanism of this phenomenon was unclear.

The concentrations of $NH_4^+\_N$, $NO_3^-\_N$, and $NO_2^-\_N$ in the effluent of the SBRs are shown in Figure 7a–c. The unit that deteriorated most severely was SBR 1. Without adding bio-accelerators or adjusting operational conditions, the $NH_4^+\_N$ concentration in the effluent of SBR 1 rose from 0.41 mg/L on day 1 to 19.2 mg/L on day 15, which corresponded to the reduction in $NH_4^{++}\_N$ removal efficiency from 98.92% to 50.83% in the same period. NO2-_N concentration of SBR 1 effluent increased slightly to 0.09 mg/L, and $NO_3^-\_N$ concentration dropped to 11.13 mg/L at the end of the recovery phase. The effluent $NH_4^+\_N$ concentration in SBR 1′, SBR 2, and SBR 3 increased from 0 mg/L on day 1 to 12.62, 8.22, and 8.16 mg/L on day 15, respectively. Correspondingly, the $NH_4^+\_N$ removal rate of those SBR units degraded from 100% to 67.67%, 78.95%, and 79.1%. For extending aeration time, adding bio-accelerators (SBR 3′) could help slightly reduce the $NH_4^+\_N$ removal rate degradation, as the $NH_4^+\_N$ removal rate in SBR 3′ dropped to 81.65% compared with 79.1% in SBR 3. The combination of stepwise increase inlet volume and adding bio-accelerators was the most effective for maintaining high nitrification efficiency in this case. The $NH_4^+\_N$ removal rate of SBR 2′ was observed to be maintained over 99.9% until day 11 before gradually dropping to 98.21% and 90.23% on the 13th and 15th recovery day, respectively. Considering SOUR of AOB (Figure 6), the inhibition rates due to the loss of system stability in SBR 1 to SBR 3′ on the ninth recovery day were 54.26%, 51.39%, 37.88%, 2.93%, 39.59%, and 25.42%, respectively, and those values on day 15 were 73.81%, 48.63%, 49.25%, 12.68%, 51.52%, and 33.02%, respectively. Hence, SBR 2′ had the highest maintenance effects on $NH_4^+\_N$ removal rate, although the effect was not the most desirable value. Hence, more research is needed to figure out the best method for preventing biological deterioration in an activated sludge system after long-term starvation.

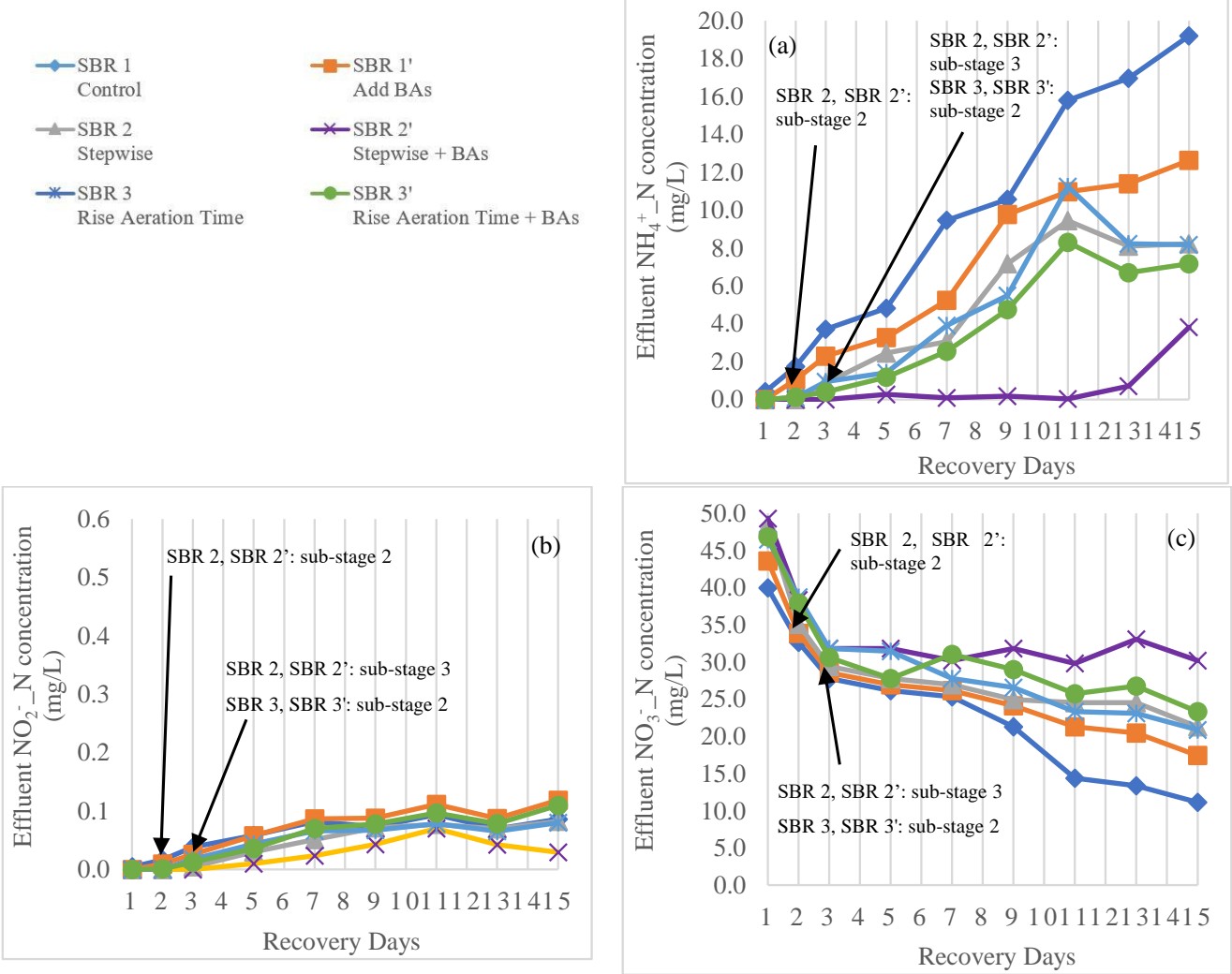

**Figure 7.** Phase 3—Effluent $NH_4^+$_N concentration (**a**), $NO_2^-$_N concentration (**b**), $NO_3^-$_N concentration (**c**).

Similar to the drop in $NH_4^+$_N removal rate, inhibition of $NO_2^-$_N oxidation rate could also be observed due to the slight increase in $NO_2^-$_N concentration in the SBRs' effluent. Although the nitrite nitrogen concentration of the SBRs was still low, mostly lower than 0.1 mg/L, it should not be neglected that the amount of $NO_2^-$_N to be oxidized in the mixed liquor of the SBRs reduced over time due to the gradual deterioration of the NH4$^+$_N removal rate. Compared with no $NO_2^-$_N found in the SBRs' effluent on day 1 and day 2, little concentration of $NO_2^-$_N was found in all of the SBRs at the end of the recovery operation period. However, it was clear that the amount of nitrite to be oxidized was not as much initially. Results of SOUR experiments (Figure 6) further confirmed the assumption, as SOUR$_{NOB}$ of SBR 1 to SBR 3′ dropped by 45.06%, 17.97%, 4.25%, 1.35%, 10.64%, and 5.08%, respectively, on day 9. Those figures then continued to drop to 29.22%, 14.44%, 26.01%, 14.31%, 17.78%, and 14.49%, correspondingly, on day 15. As are shown in Figure 7b, most nitrite nitrogen was oxidized to a nitrate; hence, a nitrite accumulation phenomenon was not observed in this phase.

It is difficult to precisely point out the reason for nitrification deterioration in this phase. Yilmaz et al. [11] stated that the transient period from a starved to a complete functional system is critical and needs to be managed closely. The levels of intracellular materials, such as enzymes and RNAs of the microorganisms, are likely to be low, even when a sufficient amount of biomass is maintained, due to maintenance mechanisms

and stressful conditions and need to be re-synthesized [23]. Some cells may need to be 'woken up' through appropriate resuscitation strategies [25]. It could also be assumed that long-time starvation had promoted the growth of carnivorous bacteria that caused deterioration in a bacterial consortium of nitrifiers. Coello Oviedo et al. [10] reported the growth of opportunist carnivorous bacteria in a starved bioreactor. This hypothesis could be an appropriate reason to explain the gradual deterioration of nitrification efficiency in the system regarding the publication of Coello Oviedo et al. [10], which described the strong development of opportunist carnivorous microorganisms after around 10 days of an aerobic starvation period and those microorganisms only found to disappear on the last days of the starvation experiment (i.e., around the 20th day). Therefore, there was a high possibility that carnivorous microorganisms had developed rapidly in the 14 days of starvation in the SBR system in this study.

Furthermore, the seed sludge in this study was highly likely to contain higher microorganisms due to regular starvation during long-term vacations. The excess predation on nitrifiers by carnivorous organisms was found to cause the deterioration of nitrification efficiency in previous research. Rotifers, one of the genera reported to harm nitrification by grasping behavior [26], developed abundantly in aerobic starvation [27]. The negative effect on predators was more severe to slow-growing autotrophic nitrifiers than fast-growing heterotrophs [26,28]. For avoiding overgrazing on nitrifiers by predators, the method of adding some predator inhibitors could be adapted. For example, Moussa et al. [4] added NaCl with the concentration of 5 gCl/L to eliminate protozoa and metazoa from activated sludge. The activity of AOB and OHOs was found to recover after washing away the salt concentration quickly. Other selective inhibitors such as cycloheximide and nystatin at a concentration of 8 mg/L each could also be used [26]. However, the effect of predation by higher microorganisms is out of the scope of this research. Further studies should be conducted to elucidate the main reason for this phenomenon to prevent future occurrences.

*3.4. Discussion*

Adjusting operational conditions to boost the recovery rate of inhibited activated sludge systems, i.e., reducing the loading rate of pollutants to put less stress on the microorganisms, hence accelerating the recovery rate, has been reported in several pieces of research. Previous studies by the authors of [29,30] confirmed that change in HRT has substantial impact on the microbial community and system efficiency. The parameter should be optimized based on the concept of balancing nutrient removals with operating costs. Wang et al. [15] also observed that the bioreactor unit that changed the aerobic time from 5 h to 9 h required 7 days less than the control group to recover $NH_4^+\_N$ removal efficiency fully. Similarly, past studies also proved that resuscitating the starved biomass in stepwise mode can reduce the time required for a full recovery. Yilmaz et al. [11] applied stepwise resuscitation in a starved SBR (30 days) with only 50% followed by 75% of the original inlet feed on the first and second day of recovery, respectively, and 100% on the fourth day (slightly similar to the strategy in this study). As a result, the nitrification rate in SBR quickly recovered to its initial value within 4 days of resuming system operation and remained stable until the next one month. Torà et al. [12] also achieved full recovery of ammonia removal within 5 days in a pilot plant using an automatic control loop modifying the applied nitrogen loading rates.

This is the first study to investigate the appropriateness of introducing bio-accelerators to boost the recovery rate of the salinity-inhibited and starved biomass system and combine previously studied methods (i.e., extending aerobic time and stepwise resuscitation of the system) with bio-accelerators addition. More desirable methods could be approached by coupling bio-accelerators addition with the latter methods, as shown in the previous section of this report. It was especially noticeable that the combination of stepwise strategy and bio-accelerator introduction in SBR 2′ showed higher efficiency in recovery, as seen in Phases 1 and 2 and maintaining treatment efficiency in Phase 3.

For boosting the recovery efficiency of the inhibited activated sludge system by merely dosing BAs, it could be seen that there was a time requirement for those BAs to take effect and significantly stimulate the recovery rate of nitrification activity. From the recovery behavior of SBR 1′ in Phases 1 and 2 of this research, the required time for BAs to take effect was 19 days for salinity shock and 6 days for anoxic–anaerobic starvation. Before this time, the effects of adding BAs toward the recovery behavior of the inhibited systems was not very considerable compared with other recovery stimulating methods. Meanwhile, the time required for BAs to significantly take effect for AOB recovery in Phase 3 was significantly shorter. The difference may come from the mechanism of L-aspartic acid, which could be more favorable for the re-growth of AOB rather than NOB.

In comparison between the effect of stepwise strategy and lengthening aeration time, specifically in boosting salinity-inhibited SBRs without considering the addition of bio-accelerators, it was the first method that had a more pronounced impact on the recovery efficiency of the SBRs, as both SBR 2 and SBR 2′ could recover to the initial treatment effectiveness. At the same time, SBR 3 may have needed additional time to reach complete nitrite oxidation. In Phase 2, meanwhile, considering the stimulating effects of three single methods—i.e., adding BAs, stepwise increase feed loading rate, and increasing aeration time—it was the method applied for SBR 3 (increase aeration time) that had a slightly higher recovery boosting effect than the other methods applied in SBR 1′ and SBR 2. However, it is important to notice that the $NH_4^+$_N and $NO_2^-$_N concentrations measured in the effluent of SBR 2 and SBR 2′ in the first recovery days of all three phases were deficient as compared with other units. The nitrification efficiency of both reactors in sub-stage 1 (400 mL feed flowrate, 1.6L of reaction volume) was highly effective; hence, both SBR 2 and SBR 2′ were able to switch to sub-stage 2 within the first 3 recovery days. In other words, the activated sludge unit could eliminate ammonium and nitrite nitrogen in the liquid phase quickly by applying this method. This is attributed to two factors achieved simultaneously: reducing loading rates of ammonium nitrogen and increasing biomass concentration as the liquid phase becomes denser. Meanwhile, the time required to extend the aeration time applied in SBR 3 and SBR 3′ to achieve the same results was longer. As mentioned above, the addition of BAs in the system required some specific time to achieve a significant effect. Moreover, adding bio-accelerator ingredients, including biotin, 6-benzylaminopurine, and L-aspartic acid, could also stabilize the nitrification rate in the system's recovery [13]. Considering these factors makes it understandable that coupling the stepwise strategy with BAs addition showed more desirable results than coupling extending aeration time with BAs addition throughout this research. However, the impacts on recovery acceleration by adopting a stepwise strategy or extending aeration time alone were nearly similar. In Phase 3, the SBR 2′ could maintain complete nitrification until day 11, which was a lot better than that observed in SBR 3′, despite the similarity in deterioration rate recorded in SBR 2 and SBR 3. To explain this phenomenon, it should be noticed that SBR 1 and SBR 1′ without adjusting the operational condition could already achieve high treatment efficiency of $NH_4^+$_N. The reason for subsequent nitrification degradation could be an unstable activity of microorganisms due to sudden system resuscitation at full loading rate after long-term starvation (1) and the development of higher microorganisms that prey on nitrifiers (2). Although longer aeration time meant a lower daily loading rate for SBR 3 and SBR 3′, this may also give more time for higher microorganisms to consume nitrifiers. For the full nitrification rate achieved in SBR 1 and SBR 1′ measured on the first recovery day, as mentioned above, it could be understood that 400 min of aeration was still enough for complete oxidation of ammonium nitrogen at the time. Meanwhile, for SBR 3 and SBR 3′, 640 min of aeration might have provided an additional time of 240 min for the activity of carnivorous microorganisms. Thus, it is understandable that the population of higher microorganisms might grow faster in SBR 3 and SBR 3′ than those in other SBRs due to this redundancy in aeration time. Although the combination of adding bio-accelerators and stepwise increased feed flowrate could show high effectiveness in maintaining system stability, as observed in SBR 2′, a denser population of carnivorous microorganisms may

be responsible for the faster nitrification deterioration rate in SBR 3′. In addition, a rapid decrease in $NH_4^+$_N removal efficiency recorded in SBR 2 and SBR 3 also indicated that the initial operational adjustment of only 2 days was not enough to 'waken up' the full activity of the microbial consortium in the system. As the growth rate of carnivorous heterotrophic bacteria is known to be much higher than that of autotrophic nitrifiers (0.45 gVSS/gCOD for heterotrophic bacteria compared with 0.1–0.15 g VSS/g $NH_4^+$-N for AOB [31]), dosages of BAs seemed to be not enough for maintaining an appropriate nitrifiers population in this case. In the recovery period of the anaerobic starvation phase, however, the detrimental phenomenon causing a severe deterioration in nitrification rate occurring in Phase 3 was not recorded. Blaming the cause of nitrification deterioration by excess predation on higher microorganisms that abundantly grow under the previous aerobic famine condition, such a phenomenon was avoided under anoxic–anaerobic starvation simulated in this phase. Lopez et al. [27] reported the absence of protozoa and rotifers under anaerobic starvation conditions while those microorganisms increased significantly under aerobic starvation. As rotifers are sorted as aerobic microorganisms, incorporation of anoxic conditions may hinder their growth and activity, hence minimizing the grazing pressure of the genus on autotrophic nitrifiers [32].

## 4. Conclusions

In this study, there were five different recovery stimulation methods, including dosing bio-accelerators, stepwise increase feed flowrate, stepwise increase feed flowrate coupled with dosing bio-accelerators, extended aeration time, and extended aeration time coupled with dosing bio-accelerators, proposed to boost the recovery process of the SBRs system. All of the recovery stimulation methods proposed in this study boosted the effectiveness of system recovery to some specific extent. The integrated method of stepwise loading strategy and dosing BAs is recommended for process recovery after high salt inhibition and anaerobic starvation. If BAs could not be prepared, a stepwise strategy should be adopted to boost the recovery rate of systems inhibited by high salt concentration. Furthermore, lengthening aeration time should be implemented to shorten the time required for complete recovery after resuscitating the system from anaerobic starvation. For process recovery after long-term aerobic starvation, perhaps inhibitors of predators should be dosed before resuming system operation. Further experiments should be conducted to confirm this hypothesis.

**Supplementary Materials:** The following supporting information can be downloaded at: https://www.mdpi.com/article/10.3390/w14010048/s1, Figure S1: Phase 2—Change of mixed liquor's characteristics in the SBRs during anaerobic starvation; Figure S2: Phase 2—Change of biomass concentration of the SBRs during anaerobic starvation; Figure S3: Phase 3—Change of mixed liquor's characteristics of the SBRs during starvation; Figure S4: Phase 3—Change of biomas characteristics of the SBRs during starvation.

**Author Contributions:** Conceptualization, H.-D.N., V.-P.L. and Y.-P.T.; resources, Y.-P.T.; writing—original draft preparation, H.-D.N.; writing—review and editing, K.-C.D. and Y.-P.T.; supervision, Y.-P.T.; project administration, C.-C.Y.; funding acquisition, Y.-P.T. All authors have read and agreed to the published version of the manuscript.

**Funding:** This study was supported by Grants received from the Ministry of Science and Technology, Taiwan, Republic of China (MOST 110-2221-E-260-011- and MOST 110-2622-E-260-002-).

**Institutional Review Board Statement:** Not applicable.

**Informed Consent Statement:** Not applicable.

**Data Availability Statement:** All data used during the study appear in the published article.

**Conflicts of Interest:** The authors declare no conflict of interest.

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
