# Peer review of "Comparative Study on Using Various Recovery Stimulation Methods to Boost Nitrification Recovery in SBRs Inhibited by Hazardous Events"

_water, doi:10.3390/w14010048_

Round 1

Reviewer 1 Report

Dear Editor,

I have carefully reviewed the manuscript titled “Comparative Study on Using Various Recovery Stimulation Methods to Boost Nitrification Recovery in SBRs Inhibited by Hazardous Events” authored by Hoang-Duy Nguyen et al.

The manuscript is interesting and the authors have obtained a lot of interesting results. Manuscript describes a system consisting of 6 SBR units in parallel for 3 phases. The investigation of the impacts of salinity shock, anaerobic and aerobic starvation on the activated sludge process stability and effects of various recovery stimulation methods on the subsequent recovery period has been carried out by the authors. Different recovery strategies were applied in each SBR unit, including natural recovery, adding bio-accelerators, stepwise increase feed strategy, stepwise strategy coupling with bio-accelerators dosing, extending aeration time, and extending aeration time coupling with bio-accelerators dosing.

I strongly recommend to accept this manuscript after minor revision.

My comments are the following:

  • Some sentences in the manuscript are so long. Please check it and provide editing.
  • The figure 1 is not quality much. Please improve the figures 1.
  • Author could discus more about the role of microbial communities in the water and wastewater treatment

https://www.mdpi.com/2227-9717/8/12/1546

https://doi.org/10.3390/biom10060921

http://dx.doi.org/10.1007/s13205-019-2041-9

Best wishes,

Reviewer

Author Response

Dear Editor

This is the revised manuscript entitled “Comparative Study on Using Various Recovery Stimulation Methods to Boost Nitrification Recovery in SBRs Inhibited by Hazardous Events”(water-1510135). It is resubmitted to consider for publication as an “original research paper” in Water. All the authors mutually agree for submitting this manuscript to Water. The point-to-point responses to reviewers and editors are shown below.

Please kindly consider it for publication in your journal.

Thanks for your great help.

Best Regards,

Yung-Pin Tsai

=============================================================

review-1

Comments and Suggestions for Authors

Dear Editor,

I have carefully reviewed the manuscript titled “Comparative Study on Using Various Recovery Stimulation Methods to Boost Nitrification Recovery in SBRs Inhibited by Hazardous Events” authored by Hoang-Duy Nguyen et al.

The manuscript is interesting and the authors have obtained a lot of interesting results. Manuscript describes a system consisting of 6 SBR units in parallel for 3 phases. The investigation of the impacts of salinity shock, anaerobic and aerobic starvation on the activated sludge process stability and effects of various recovery stimulation methods on the subsequent recovery period has been carried out by the authors. Different recovery strategies were applied in each SBR unit, including natural recovery, adding bio-accelerators, stepwise increase feed strategy, stepwise strategy coupling with bio-accelerators dosing, extending aeration time, and extending aeration time coupling with bio-accelerators dosing.

I strongly recommend to accept this manuscript after minor revision.

My comments are the following:

  • Some sentences in the manuscript are so long. Please check it and provide editing.

Thanks a lot for reviewer’s comment. The manuscript has been fastidiously revised again by the authors.

  • The figure 1 is not quality much. Please improve the figures 1.

Thanks a lot for reviewer’s comment. The Figure 1 has been replaced by an improved version.

  • Author could discuss more about the role of microbial communities in the water and wastewater treatment

Thanks a lot for reviewer’s comment. It’s really difficult for us to discuss the results according to individual microbial communities due to lack of relating data. We collected the MLVSS and MLSS data to compare the variation of the whole microbial communities and we also collected SOUR data to show the variation of the whole microbial activities. Both of them (MLVSS/MLSS and SOUR) were representation of microorganisms in activated sludge and have been thoroughly discuss in the manuscript.

Reviewer 2 Report

The paper discusses different scenarios of nitrification malfunctions and possible operational solutions along with the effect of adding bio-accelerators to recover nitrification rate. The methodology is well designed and the results are presented in a fairly clear manner though there are parts that would need further elucidation.

These are mentioned in the order of their occurrence along with other minor comments:

It is slightly more elegant to mention each source separately. Examples for lumped references: 4-7, 13-15. Whether these are modified or not, is left to the authors' discretion. 

Lockdown is mentioned as a reason for starvation in bioreactors - could the authors give an example to what kind of wastewater treatment plant would suffer in such a manner? Later the wwtp of university campus is mentioned which probably had this issue when students went into online studying, but it is not common to have separate biological treatment systems for institutions in urban areas, so the reader might ponder until they come to the conclusion.

On page 3 line 92, the full expression should be mentioned first then the abbreviation in brackets .

For the salt shock it is mentioned that salt was added  in3 consecutive phases, did the authors mean cycles? I.e., did it last for 24 hours? Isn't that a little too short period compared to the starvation periods? Maybe a short explanation on the time periods should be included.

In line 159, the name of the method should also be mentioned before the source.

Can the authors please quantify the amount of alkalinity instead of saying "some". 

In which cycle and at which stage were the samples taken? When did the first cycle start in a day? Could these information included in the methods section?

In lines 250, 268, the sentences should be revised, it seems that  some expressions are missing. 

The figure editing requires revision, some of them don't show the labels properly, others miss axis notations and the days should be under each other. Coherent look help the comprehension and the overall judgment of the manuscript.

It wasn't fully clear how and when and how often the bio-accelerators were added to the system/influent. Please, specify it.

The growth rate of AOB and NOB are mentioned several times but they are not quantified, nor is a source provided for the information. Please include concrete values with sources.

Supplemental materials were not included in the downloadable manuscript so the reviewer could not check those results. 

The style of citation is different - Anthonisen et al., 1976 is used instead of numbering.

In which tank did the authors experience pH drop?

How does the ammonium oxidation connect to the anaerobic starvation? The authors probably wanted to refer to the first period of recovery but it isn't clear from the text.

Since data from the starvation period is not included, it was a little confusing to read the results from the diagrams. The text for the anaerobic starvation might need a little bit of clarification, the style is now a little bit rushed, a lot has to figured out by the reader as the data in the text and the charts are deciphered. This is just a style suggestion for better reading experience.

Same goes for the aerobic starvation section. Why do the authors start with the SOUR results instead of following the order they used earlier? This and the mention of having good nitrification rate in the first period thus quickly changing to the second phase was somewhat confusing. Sentences such as "The precise reason or mechanism of this phenomenon was unclear." did not help either. Maybe together with the information in the supplementary material, it would be clearer but currently the text needs some elucidation.

In general, it was interesting to read about the findings of this article.

Author Response

review-2

Comments and Suggestions for Authors

The paper discusses different scenarios of nitrification malfunctions and possible operational solutions along with the effect of adding bio-accelerators to recover nitrification rate. The methodology is well designed and the results are presented in a fairly clear manner though there are parts that would need further elucidation.

These are mentioned in the order of their occurrence along with other minor comments:

It is slightly more elegant to mention each source separately. Examples for lumped references: 4-7, 13-15. Whether these are modified or not, is left to the authors' discretion. 

Thanks a lot for reviewer’s comment. We respect everyone’s point of view. However in this case we decided to keep this style of references due to the limited time for revision from editor request of this journal.

Lockdown is mentioned as a reason for starvation in bioreactors - could the authors give an example to what kind of wastewater treatment plant would suffer in such a manner? Later the wwtp of university campus is mentioned which probably had this issue when students went into online studying, but it is not common to have separate biological treatment systems for institutions in urban areas, so the reader might ponder until they come to the conclusion.

Thanks a lot for reviewer’s comment. Any WWTP could be suffered from lockdown situation. For industrial WWTP, the operator may have to provide carbon and nutrient source for maintaining the microbial activities of the biological unit. In lockdown situation it is very hard for the operators to maintain this activity. There is another case that the WWTP might face to starvation in bioreactors when earthquake occurs and the influent main pipe is broken down. There also is the same situation in a University Campus during summer vacation.

On page 3 line 92, the full expression should be mentioned first then the abbreviation in brackets .

Thanks a lot for reviewer’s comment. The error has been corrected in the manuscript.

For the salt shock it is mentioned that salt was added in 3 consecutive phases, did the authors mean cycles? I.e., did it last for 24 hours? Isn't that a little too short period compared to the starvation periods? Maybe a short explanation on the time periods should be included.

Thanks a lot for reviewer’s comment. We agree that the salt shock period is too short compared to starvation. In this case we regarded NaCl as a toxic compound which may pose effects as soon as it is added, and this is not the same for starvation. In this research, moreover, we aimed at the recovery ability of the biomass rather that their ‘adaptability’. Hence, we think that 24 hours is barely enough, this range of time is also similar to the report of Yogalakshmi et al. [19].

In line 159, the name of the method should also be mentioned before the source.

Thanks a lot for reviewer’s comment. We have put the name of the method before the source as recommended.

Can the authors please quantify the amount of alkalinity instead of saying "some". 

Thanks a lot for reviewer’s comment. While measuring the SOUR of the microorganisms we added drops of 0.1 NaOH to keep pH value at 7.5 ± 0.2. Also, we also noticed that the word “some” may come up as a bit shady. So we have put more details in that sentence.

In which cycle and at which stage were the samples taken? When did the first cycle start in a day? Could these information included in the methods section?

Thanks a lot for reviewer’s comment. As we operate the SBRs for 3 cycles per day and 8 hours for each cycle, the cycle time frame are 7:00 – 15:00 (cycle 1), 15:00 – 23:00 (cycle 2) and 23:00 to 7:00 (cycle 3). For SBR 3 and SBR 3’ during the 2-cycle-per-day period, the time for 2 cycle was 07:00-19:00 (cycle 1) and 19:00 – 07:00 (cycle 2). The samples were taken at the end of cycle 1, representing the treatment efficiency of that operation day.

In lines 250, 268, the sentences should be revised, it seems that some expressions are missing. 

Thanks a lot for reviewer’s comment. For line 250 we think that the phrase may cause a bit confusion to the reader and we slightly modified it a little bit to make it more comprehensible. Similarly, the phrases mentioning the time requirement for effects from BAs in line 268 was also removed as this detail will be mentioned later (line 540 to 542).

The figure editing requires revision, some of them don't show the labels properly, others miss axis notations and the days should be under each other. Coherent look help the comprehension and the overall judgment of the manuscript.

Thanks a lot for reviewer’s advice. The authors have revised and fixed the figures.

It wasn't fully clear how and when and how often the bio-accelerators were added to the system/influent. Please, specify it.

Thanks a lot for reviewer’s comment. The information was mentioned from line 151 to 153. We manually added 1 ml of the stock solutions directly into the SBR 1’, SBR 2’ and SBR 3’ right after the influent filling have been finished (i.e. when the aeration step begin). The action was continuously repeated in every cycle during the whole recovery process. We hope the change in the manuscript help clarifying this idea.

The growth rate of AOB and NOB are mentioned several times but they are not quantified, nor is a source provided for the information. Please include concrete values with sources.

Thanks a lot for reviewer’s comment. We have added growth rates for the microorganisms with source in the manuscript.

Supplemental materials were not included in the downloadable manuscript so the reviewer could not check those results. 

Thanks a lot for reviewer’s comment. We will provide the file for supplemental data along with this revised manuscript.

The style of citation is different - Anthonisen et al., 1976 is used instead of numbering.

Thanks a lot for reviewer’s comment. We have revised and corrected that mistake.

In which tank did the authors experience pH drop?

Thanks a lot for reviewer’s comment. Are you mentioning the drop of pH discussed from line 340 to 346? We measured the drop of pH in all of the SBRs after just 1 day of anaerobic starvation. The phenomenon was illustrated in Figure 1 of the Supplemental Data file which we will provide together with this revised manuscript.

How does the ammonium oxidation connect to the anaerobic starvation? The authors probably wanted to refer to the first period of recovery but it isn't clear from the text.

Thanks a lot for reviewer’s comment. In the period of anaerobic starvation, ammonium oxidation did not happen hence there was accumulation of NH4+-N recorded in the mixed liquor. These results are included in Supplemental Data Figure 1 which will be attached to this revised manuscript.

Since data from the starvation period is not included, it was a little confusing to read the results from the diagrams. The text for the anaerobic starvation might need a little bit of clarification, the style is now a little bit rushed, a lot has to figured out by the reader as the data in the text and the charts are deciphered. This is just a style suggestion for better reading experience.

Thanks a lot for reviewer’s comment. The authors have checked the related paragraphs and made modifications for better comprehension.

Same goes for the aerobic starvation section. Why do the authors start with the SOUR results instead of following the order they used earlier? This and the mention of having good nitrification rate in the first period thus quickly changing to the second phase was somewhat confusing. Sentences such as "The precise reason or mechanism of this phenomenon was unclear." did not help either. Maybe together with the information in the supplementary material, it would be clearer but currently the text needs some elucidation.

Thanks a lot for reviewer’s comment. The authors have checked the related paragraphs and made modifications for better comprehension. For the text "The precise reason or mechanism of this phenomenon was unclear.", we tried to put some explanation based on other papers but as we don’t have any data to completely defend our ideas. This was also an unexpected result and this is not included in the purpose of this study. We just wanted to emphasize this unexpected result to readers and give them reference in their future research.

In general, it was interesting to read about the findings of this article.
